# Development of Triangle RNA Nanostructure for Enhancing RNAi-Mediated Control of *Botrytis cinerea* Through Spray-Induced Gene Silencing Without Extra Nanocarrier

**DOI:** 10.3390/biology14111616

**Published:** 2025-11-18

**Authors:** Ya Chen, Yiqing Liu, Yani Huang, Fangli Wu, Weibo Jin

**Affiliations:** 1Key Laboratory of Plant Secondary Metabolism and Regulation of Zhejiang Province, College of Life Sciences and Medicine, Zhejiang Sci-Tech University, Hangzhou 310018, China; 2Zhejiang Sci-Tech University Shaoxing Academy of Biomedicine, Shaoxing 312366, China

**Keywords:** *Botrytis cinerea*, SIGS, dsRNA, RNA nanotechnology, RNA nanoparticles

## Abstract

Gray mold, caused by the fungus, *Botrytis cinerea*, damages over 1400 plant species and leads to serious crop losses worldwide. People often use chemical fungicides, but these will harm the environment and may lose effectiveness after using for long time. As a safer alternative, we developed a tiny triangular RNA particle, called “Bc-triangle,” that silence fungal genes needed for growth and infection. When sprayed on plants, Bc-triangle was more stable and effective than regular RNA, protecting crops for up to 10 days. Tests showed it slowed fungal growth, suppress disease lesion area expansion on leaves, and strongly suppressed the fungus’s target genes. This study highlights RNA nanotechnology as a sustainable and chemical-free strategy for crop protection, with potential benefits for food security and environmental health.

## 1. Introduction

*Botrytis cinerea*, the causative agent of gray mold, is a globally prevalent necrotrophic fungal pathogen with an exceptionally broad host range, encompassing vegetables, fruits, ornamental plants, and trees, affecting up to 1400 plant species [1,2]. Annual preharvest losses attributed to *B. cinerea* are estimated at 10–30%, with postharvest rot rates reaching up to 50%, resulting in over $10 billion in global economic losses [3,4]. The control of *B. cinerea* remains a major challenge due to its broad host range, rapid reproduction, and strong adaptability [5].

Conventional management strategies predominantly depend on chemical disinfectants. However, their extensive utilization has resulted in the emergence of resistance, heightened environmental concerns, and the intensification of food safety concerns. Consequently, the priority is to cultivate sustainable, precise, and environmentally friendly alternatives.

RNA interference (RNAi) has emerged as a novel approach for plant disease control through gene-specific silencing in pathogens. Spray-induced gene silencing (SIGS) [6,7], which involves the exogenous application of double-stranded RNA (dsRNA) onto plant surfaces, enables transient silencing of essential fungal genes without the need for transgenic plants [8,9]. Despite its promise, SIGS is often limited by the instability of naked dsRNA under field conditions, inefficient uptake by fungal cells, and short protection windows, all of which restrict its practical application [10].

Recent advances in nanotechnology offer new avenues to address these challenges. Nanocarriers such as layered double hydroxides (LDHs), chitosan complexes, and carbon-based materials have been explored to enhance the stability and cellular uptake of dsRNA [11,12,13]. While effective, most of these carriers rely on synthetic or exogenous materials, posing scalability, biocompatibility, and environmental risks [8,13].

Structured RNA nanoparticles—constructed using RNA origami techniques—offer precise spatial arrangement, increased nuclease resistance, and enhanced cellular uptake [8]. In this study, we designed and constructed a triangular RNA nanoparticle (termed Bc-triangle) incorporating siRNAs targeting four key virulence genes of *B. cinerea*—*BcDCL1*, *BcPPI10*, *BcNMT1* and *BcBAC*. We hypothesized that this multivalent nanostructure could enhance RNAi efficiency and provide superior antifungal effects compared to conventional dsRNA [14]. The efficacy of the Bc-triangle was evaluated in vitro and in planta, and its gene-silencing performance was confirmed via quantitative PCR analysis. Our results demonstrate that RNA nanostructures targeting virulence genes can significantly enhance the effectiveness of SIGS, offering a promising new strategy for fungal disease management in crops.

In this study, we selected a square RNA nanostructure, as reported by Li et al. [15], to design an antifungal RNA fungicide. This was achieved by linking multiple siRNAs targeting four virulence genes Bcin12g06230.1 (Dicer-Like 1, *DCL1*), Bcin13p01840.1 (Peptidyl prolyl cis-trans isomerase 10, *PPI10*), Bcin04p04920.1 (N-myristoyltransferase 1, *NMT1*) and Bcin15p02590.2 (putative adenylate cyclase, *BAC*)) which was validated as an effective target gene for the RNAi-based management of Botrytis cinerea [16,17].

## 2. Materials and Methods

### 2.1. Preparation of Plant Materials and Botrytis cinerea

Tobacco plants (*Nicotiana benthamiana*) were cultivated in a greenhouse, with an approximate temperature of 26 °C and a photoperiod of 16 h of light and 8 h of darkness. The fungus *Botrytis cinerea* was cultivated on potato dextrose agar (PDA) medium at 26 °C with 80% humidity. Following a two-week cultivation period, conidia were harvested from the PDA cultures and suspended in sterile deionized water to achieve a final concentration of 5 × 10^6^ conidia/mL for subsequent bioassays.

### 2.2. Design of RNA Nanoparticles

The gene-specific small interfering RNAs (siRNAs) were designed using the Designer of Small Interfering RNA (DSIR, CEA, Paris, France) algorithm, with selection criteria including a prediction score above 90 and the absence of more than three consecutive identical nucleotides [18]. Potential off-target effects were assessed by performing NCBI BLASTn (version 2.12.0) searches of the siRNA sequences against the tomato genome, using default parameters with the exception of a word size set to 7 [19]. After eliminating off-target siRNAs, five siRNAs were randomly selected to design. A triangular RNA nanoparticle (Bc-triangle) was designed by integrating the five selected siRNAs using tetra-U helix linking motifs [15], hairpin loops (5′-UCCG-3′), and kissing loops (5′-AAGGAGGCA-3′, 5′-AAGCCTCCA-3′), resulting in the formation of a triangle RNA nanostructure. The secondary structure of the RNA square was confirmed using the RNAfold program [20]. In addition, all of these six siRNAs targeting DCL1, PPI10, Nmt1 and BAC were concatenated to form a long double-strand RNA (Bc-dsRNA) as the positive control.

### 2.3. Plasmid Construction and RNA Nanoparticle Expression in E. coli

The coding sequence of Bc-triangle was subjected to a chemical synthesis process and subsequently cloned into the *pET-28a* (*+*) expression vector, a maneuver that utilized the restriction sites Xba I and Bpu1102 I (GenScript, Nanjing, China). The recombinant plasmid was then transformed into the *Escherichia coli* HT115 (DE3) strain, and RNA expression was induced at 37 °C with 1 mM isopropyl β-D-1-thiogalactoside (IPTG) for a period of 4 h. The kit utilized for dsRNA extraction (RNA-Direct Silencing Technology Co., Ltd., AK011, Shaoxing, China) was employed in accordance with the protocol delineated in the kit manual, and then subjected to agarose gel electrophoresis to verify the production of RNA nanoparticle.

In addition, the reverse complementary sequence of T7 promoter was first appended to the 3′-end of the coding sequence of Bc-dsRNA. This sequence was then chemically synthesized and also inserted into pET28a using Xba I and Bpu1102 I restriction endonucleases. Finally, Bc-dsRNA were performed following the same procedure as Bc-triangle.

### 2.4. Spore Germination Assay

Conidial suspensions of *B. cinerea* (1 × 10^6^ spores/mL) were incubated with either No-RNA (water was used as the negative control), Bc-dsRNA, or Bc-triangle (final concentration 100 ng/μL). After 24 h incubation at 26 °C on a glass slide, spore germination was observed under a microscope.

### 2.5. Mycelial Growth Inhibition Assay

Following the sterilization of PDA (121 °C, 15 min), the medium should be cooled to 50–60 °C, and the Bc-triangle and Bc-dsRNA should be incorporated into the PDA medium. In order to achieve the desired concentrations, 7.5 mL of RNA solution was added to each 150 mL of PDA medium, resulting in final working concentrations of 25, 50, 100, and 200 ng/μL. The control group, which did not receive any treatment, received an equal volume of ddH2O. Mycelial plugs (5 mm) from actively growing *B. cinerea* cultures were placed at the center of each plate and incubated at 26 °C for 5 days. Colony diameters were measured, and inhibition rates were calculated relative to the No-RNA controls.

For mycelial growth assays, PDA plates supplemented with RNA formulations (25–200 ng µL^−1^) were inoculated with 5 mm mycelial plugs from the growing edge of *B. cinerea* cultures. Colony diameter was measured after 5 days of incubation at 25 °C, and inhibition rate was calculated as follows:(1)$$\mathrm{Inhibition}\;(\%)=\frac{\left(C-T\right)}{C}\times 100$$
where *C* is the colony diameter of the control, and *T* is the colony diameter of the treatment.

### 2.6. In Planta Assay on Tobacco

Four-week-old *Nicotiana benthamiana* plants were sprayed with 5 mL of either sterile water, Bc-dsRNA, or Bc-triangle (200 ng/μL). At 3, 7, and 10 days post spraying (dps), individual leaves were inoculated with *B. cinerea* mycelial plugs (1 per leaf). Disease progression was assessed after 3 days by measuring necrotic lesion areas by ImageJ 1.53k (NIH, Bethesda, MD, USA). Data were analyzed from three biological replicates.

### 2.7. Quantitative RT-PCR Analysis

Mycelia from treated PDA plates were collected for RNA extraction using TRIzol reagent (Solarbio, Beijing, China). cDNA synthesis was performed using a reverse transcription kit (Takara, Dalian, China). Quantitative real-time PCR (qRT-PCR) was conducted using gene-specific primers for *BcDCL1*, *BcNMT1* and *BcBAC* with a standard SYBR Green system (Vazyme, Nanjing, China). Gene expression level was normalized to an internal reference gene, and relative expression was calculated using the 2^−ΔΔCt^ method [21].

### 2.8. Statistics Analysis

All experiments were conducted with at least three independent biological replicates. Data were analyzed using one-way ANOVA followed by Tukey’s multiple comparison test in GraphPad Prism 9.0 (GraphPad Software, Boston, MA, USA). Results are expressed as mean ± standard deviation (SD). Differences were considered statistically significant at *p* < 0.05.

### 2.9. Cryo-EM Imaging

RNA NPs was transcribed in vitro using the expression vectors and the T7 RiboMAX™ Express Large Scale RNA Production System (Gatan, Pleasanton, CA, USA) following the manufacturer’s protocol. The RNA NPs was purified by DNase I digestion for 15 min at 37 °C, followed by lithium chloride precipitation, and then melted into Tris-Mg buffer (10 mM Tris-HCl, 10 mM Mg(OAc)_2_ at a concentration of 3 μM. The RNA nanostructures were confirmed using cryo-EM following the description provided by Li et al. [15].

## 3. Results

### 3.1. Design and Production of B. cinerea-Targeted Triangle RNA Nanoparticles

To construct a *B. cinerea*-targeted RNA nanostructure, we selected four virulence genes, *DCL1*, *PPI10*, *Nmt1* and *BAC*, which have been verified as effective target genes for RNAi-based management of *B. cinerea* [16]. To design *B. cinerea*-targeted siRNAs, we utilized DSIR and randomly screened out six siRNAs for these four target genes (Table 1). These six siRNAs served as structural motifs for the assembly of triangle RNA NPs utilizing hairpin loops, kissing loops, and the tetra-U motif. Computational prediction using RNAfold confirmed that the Bc-triangle adopts a thermodynamically stable conformation with a free energy of approximately −340 kcal mol^−1^, suggesting strong intramolecular interactions and folding efficiency (Figure 1a). This assembly was validated using the RNAfold program and named Bc-triangle (Figure 1b). The coding DNA sequence of the Bc-triangle is provided in Table 1. The nanostructure of the Bc-triangle was characterized using cryo-EM. Results showed that this NP could self-assemble into the expected structures (Appendix A).

### 3.2. Production of the Triangle RNA Nanoparticles

The coding cassette was cloned into the pET-28a (+) vector and expressed in *Escherichia coli* strain *HT115* (DE3). Following IPTG induction, total RNA was extracted and analyzed by agarose gel electrophoresis. A distinct RNA band of approximately 250–500 bp was detected in bacterial lysates harboring the Bc-triangle construct, whereas no such band was observed in the empty vector control (Figure 2). These results confirmed the successful expression and accumulation of the designed RNA nanostructure in *E. coli*.

### 3.3. Bc-Triangle Strongly Inhibits B. cinerea Spore Germination

To assess the antifungal efficacy of Bc-triangle during the early stages of fungal development, its effect on *Botrytis cinerea* conidial germination was investigated. After 24 h of incubation, spores in the water-treated control group (No-RNA) germinated normally, while those treated with dsRNA exhibited partial inhibition of germination. Notably, spores treated with Bc-triangle exhibited a near-complete inhibition of germination, with negligible germ tube formation (Figure 3). These results suggest that Bc-triangle exerts potent inhibitory activity at the initial infection stage, outperforming traditional dsRNA.

### 3.4. Suppression of Mycelial Growth by Bc-Triangle

The antifungal efficacy of Bc-triangle against vegetative hyphal expansion was tested on PDA medium supplemented with the final concentrations of 25, 50, 100 ng/μL and 200 ng/μL. After 5 days of incubation, Bc-triangle exhibited a concentration-dependent inhibition of mycelial growth (Figure 4a).

At 100 ng/μL, Bc-triangle achieved an average inhibition rate of 52.35%, significantly higher than the 20.19% observed for dsRNA. At 200 ng/μL, Bc-triangle suppressed hyphal growth by 95.02%, whereas dsRNA achieved 87.31% inhibition (Figure 4b).

These findings indicate that while both RNA treatments are effective at high concentrations, Bc-triangle exhibits significantly greater antifungal potency at equivalent doses, likely due to its enhanced stability and improved cellular uptake of the nanoparticle structure.

### 3.5. Bc-Triangle Confers Enhanced Protection Against B. cinerea Infection in Planta

To evaluate the efficacy of Bc-triangle under more realistic conditions, we conducted infection assays on whole tobacco (*Nicotiana benthamiana*) plants. Plants were foliar-sprayed with Bc-triangle, dsRNA, or water (negative control) at 200 ng/μL followed by challenged with *B. cinerea* mycelial plugs at 3, 7, and 10 days post-spraying (dps). Lesion area measurements revealed that Bc-triangle conferred more robust and sustained protection against fungal infection compared to dsRNA (Figure 5a,b).

At 3 dpt, lesion areas on Bc-triangle–treated leaves averaged 0.4–0.6 cm^2^, significantly smaller than those in dsRNA-treated (1.3–1.5 cm^2^) and control plants (2.0–2.3 cm^2^) (Figure 4a). Even at 10 dpt, Bc-triangle maintained strong protection, with lesion areas remaining below 1.3 cm^2^, while dsRNA and control treatments showed extensive tissue maceration (2.3–2.7 cm^2^) (Figure 4b).

These results indicate that Bc-triangle provides extended protection compared with conventional dsRNA, prolonging the effective disease-suppression window to at least 10 days after spraying.

### 3.6. Bc-Triangle Enhances Gene Silencing Efficiency in B. cinerea

To confirm whether the antifungal effect of Bc-triangle was associated through RNAi-mediated gene silencing, three genes (*DCL1*, *NMT1* and *BcBAC*) of the four target genes were selected to detect their expression levels using qRT-PCR. Results showed that both genes were significantly downregulated in Bc-triangle treatment groups compared to the No-RNA group. However, with regard to the *BcPPI10* gene, it cannot be discounted that fluctuations in gene expression levels may occur during the course of the gray mold growth cycle. Notably, Bc-triangle induced stronger gene silencing than dsRNA (Figure 6). These findings confirm that Bc-triangle more effectively delivers siRNA payloads into fungal cells and triggers efficient gene silencing.

## 4. Discussion

In this study, we successfully developed a triangular RNA nanoparticle, termed Bc-triangle, that targets *Botrytis cinerea* virulence genes—*BcDCL1*, *BcPPI10*, *BcNMT1* and *BcBAC*—identified in our previous work [17]. The Bc-triangle nanostructure demonstrated superior antifungal efficacy compared to conventional naked dsRNA across all tested parameters, including inhibition of spore germination, suppression of hyphal expansion, reduction in plant lesion size, and enhancement of gene silencing efficiency [8,17]. However, it was not possible to detect *PPI10* at any expression level. The underlying reason for the observed absence of *PPI10* expression could be attributed to the specific expression of this gene during the course of pathogen-plant interactions. Consequently, in the course of conducting experiments on mycelium in the presence of RNA nanoparticles, the expression of the mycelium could not be detected.

The dramatic reduction in spore germination upon Bc-triangle treatment suggests efficient early-stage RNAi interference. It is known that germinating conidia are highly sensitive to environmental signals, and rapid siRNA uptake during this stage may interfere with initial infection processes such as appressorium formation or adhesion [22,23]. The enhanced performance of Bc-triangle compared to dsRNA also supports previous findings that RNA nanoparticles can improve intracellular delivery and retention [24,25]. The observed dose-dependent inhibition of mycelial growth further illustrates the functional robustness of the nanostructure. At 200 ng/μL, the Bc-triangle inhibited over 95% of fungal colony expansion—significantly higher than dsRNA at the same concentration—highlighting improved bioavailability and efficiency of gene silencing in fungal tissues.

One of the most notable outcomes is the long-lasting disease protection achieved by a single foliar spray of Bc-triangle. While the protective effect of naked dsRNA significantly diminished after 7 days, Bc-triangle maintained significant antifungal activity up to 10 days post-spraying. This extended protective window is critical for field applications where environmental factors such as rain, UV radiation, and nucleases rapidly degrade dsRNA [26,27,28]. The compact and folded architecture of the RNA nanoparticle likely confers resistance to degradation and facilitates gradual siRNA release, consistent with similar observations in biomedical RNA delivery studies [25,29].

When compared to synthetic/organic carriers (e.g., layered double hydroxide clay sLDH or chitosan nanoparticles), pure RNA nanoparticles have been demonstrated to exhibit superior stability and RNAi persistence in terms of efficacy, with a significant inhibition of spore germination and mycelial growth [8]. In terms of cost, this process can utilize microorganisms (such as *E. coli*) for scale-up, thereby eliminating preparation steps involving carrier materials and complexation, significantly reducing costs. With regard to environmental compatibility, pure RNA nanoparticles are fully biodegradable and pose no risk of carrier residue, whereas synthetic carriers may lead to non-target accumulation or ecotoxicity concerns [30]. In comparison to alternative control methodologies, chemical pesticides (e.g., carbendazim) have been observed to readily induce resistance and environmental contamination. Furthermore, genetically modified RNAi crops encounter regulatory challenges in obtaining approval for genetic modification [10]. Conversely, SIGS technology does not necessitate genetic modification and permits flexible application. Additionally, biological agents (e.g., antagonistic bacteria) demonstrate diminished efficacy under environmental constraints, whereas RNA nanoparticles exhibit enhanced stability. In this study, the protective window of Bc-triangle was observed to last up to 10 days, while naked dsRNA was found to last 3–7 days, though this was slightly shorter than the carrier-complexed dsRNA (e.g., sLDH-dsRNA was found to reach 20 days) [8,9,30].

## 5. Conclusions

Our results provide strong evidence that RNA origami-based nanostructures can significantly improve the efficacy and durability of SIGS-mediated fungal disease control. The Bc-triangle RNA nanoparticle not only enables efficient multigene silencing but also enhances RNA stability and in planta retention, overcoming critical limitations of conventional dsRNA. This study advances the development of self-delivering, environmentally friendly RNA biopesticides and opens new avenues for applying RNA nanotechnology in sustainable agriculture. Further work is needed to elucidate the uptake and translocation mechanisms of RNA nanostructures in both host plants and pathogens and to assess their long-term performance under field conditions.

## Figures and Tables

**Figure 1 biology-14-01616-f001:**
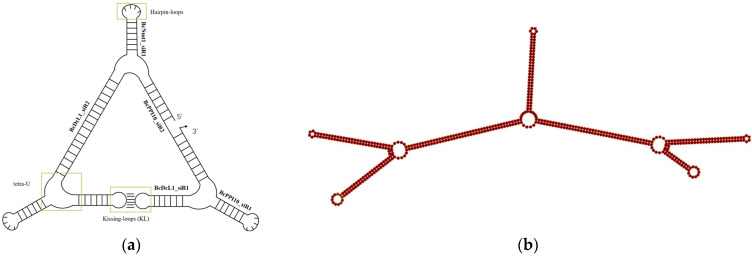
Design model of a triangle RNA nanoparticle (**a**) and its secondary structure confirmed by RNAfold (**b**).

**Figure 2 biology-14-01616-f002:**
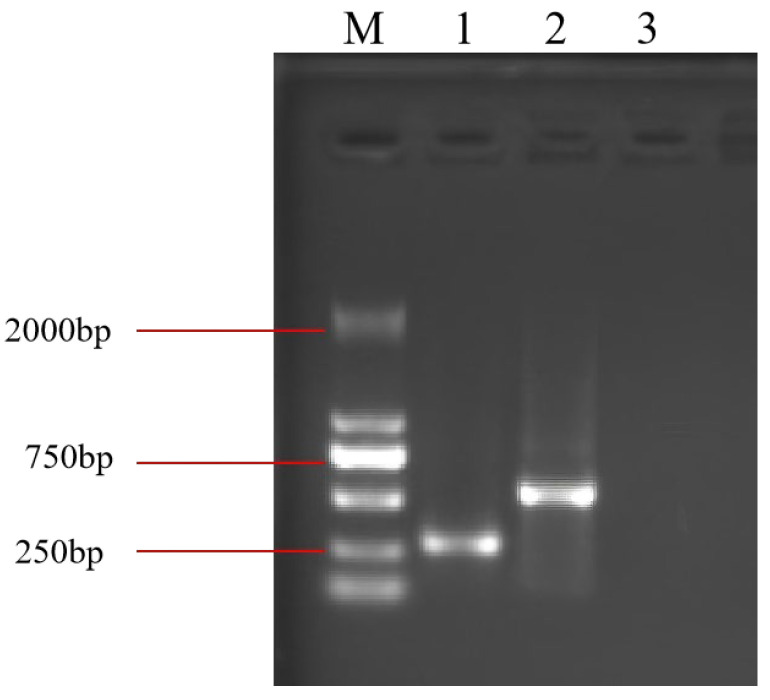
RNA extracted from induced *E. coli* harboring the empty plasmid or from the two recombinant plasmids. M: DL 2000 marker; 1: Bc-triangle; 2: Bc-dsRNA; 3: pET-28a.

**Figure 3 biology-14-01616-f003:**

Effect of Bc-triangle with concentration of 100 ng/uL for 10 h on spore germination of *Botrytis cinerea.* No-RNA served as negative control.

**Figure 4 biology-14-01616-f004:**
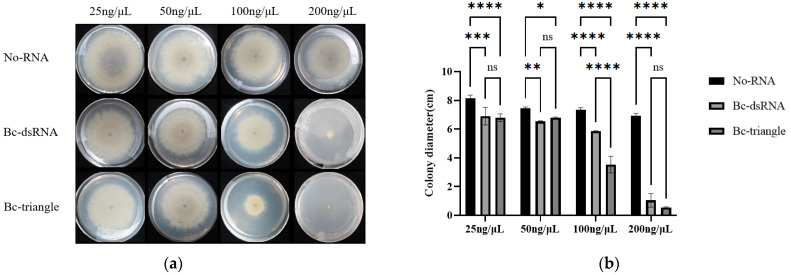
Effect of Bc-triangle with different concentrations on the growth of *Botrytis cinerea* mycelium on PDA plate. (**a**) Colony size of *B. cinerea* on PDA plates about control, dsRNAs and Bc-triangle groups. (**b**) The relative colony size of *B. cinerea*. Results are expressed as the means ± SEM of four biological replicates. Asterisk indicates the statistically significant differences according to one-way analysis of variance with Tukey test (ns: *p* > 0.05; *: *p* < 0.05; **: *p* < 0.01; ***: *p* < 0.001; ****: *p* < 0.0001).

**Figure 5 biology-14-01616-f005:**
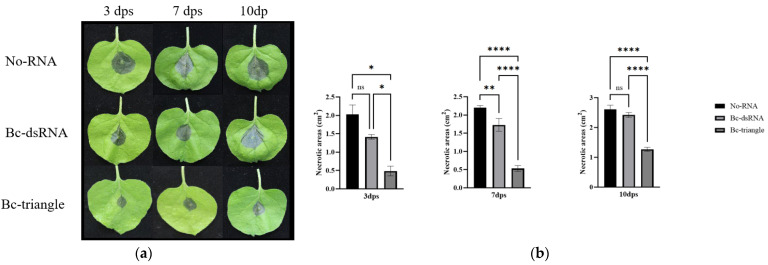
Protective efficacy of the Bc-triangle against *B. cinerea* infection on tobacco plants. (**a**) Necrosis of *B. cinerea* on tobacco leaves at 3, 7 and 10 dps. (**b**) The necrotic areas on tobacco leaves at 3, 7 and 10 dps. The results are expressed as the means ± SEMs of three leaves. Asterisk indicates the statistically significant differences according to one-way analysis of variance with Tukey test (ns: *p* > 0.05; *: *p* < 0.05; **: *p* < 0.01; ****: *p* < 0.0001).

**Figure 6 biology-14-01616-f006:**
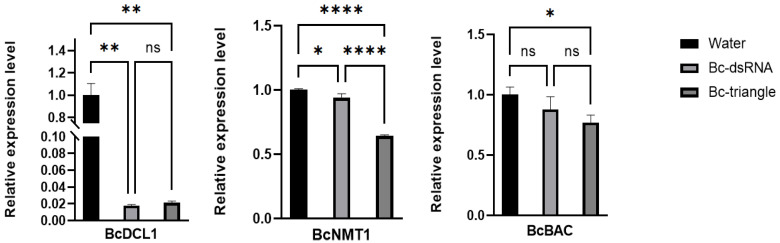
Expression levels of two target genes on *B. cinerea* treated by Bc-dsRNA and Bc-triangle. The Actin gene of *B. cinerea* was used as the internal control. The results are expressed as the means ± SEM of three biological replicates. Asterisk indicates the statistically significant differences according to one-way analysis of variance with Tukey test (ns: *p* > 0.05; *: *p* < 0.05; **: *p* < 0.01; ****: *p* < 0.0001.

**Table 1 biology-14-01616-t001:** siRNA used in RNA NP.

siRNA-Name	siRNA Sequence	Bc-Triangle
*BcDCL1*_siR1	ATTATCTCTAGCATCTACCGG	GGTAGATGCTAGAGACTTTTGCCTTGAACATCAGAACGAGGGCGACCCGCAAGGGTCGCCCTCGTTCTGATGTTCAAGGCTTTTGGCGCCGCATCTCGATGGTCAGAATTCACCGGTAGATGCTAGAGACTTTTGCCTAGATGAACAAGTATGCCGGATGGCGCAAGCCATCCGGCATACTTGTTCATCTAGGCTTTTGGGACACGACGATGAGAATAAGACATCAATGACCTGATGATGTCTTATTCTCATCGTCGTGTCCCTTTTGTCTCTAGCATCTACCGGTGAATTCTGACCATCGAGATGCGGCGCCTTTTGTCTCTAGCATCTACCGGTGAATTATCTCTAGCATCTACCGGTGACTTTTGTCTCTAGCATCATGACCTGAGATGCTAGAGACTTTTGCCATTCTTCCTATCCACCTCCAAAGCCGCAAGGCTTTGGAGGTGGATAGGAAGAATGGCTTTTGTCACCGGTAGATGCTAGAGATAATTCAC
*BcDCL1*_siR2	TAGATGAACAAGTATGCCGGA	
*BcPPI10*_siR1	ATTCTTCCTATCCACCTCCAA	
*BcPPI10*_siR2	ATTCTGACCATCGAGATGCGG	
*BcNMT1*_siR1	TTGAACATCAGAACGAGGGCG	
*BcBAC*_siR1	TGTCTTATTCTCATCGTCGTG	

## Data Availability

Data is contained within the article or Appendix A. The original contributions presented in this study are included in the article/Appendix A. Further inquiries can be directed to the corresponding author.

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
