# Peer review of "Development of Triangle RNA Nanostructure for Enhancing RNAi-Mediated Control of Botrytis cinerea Through Spray-Induced Gene Silencing Without Extra Nanocarrier"

_biology, 2025, doi:10.3390/biology14111616_

Round 1
Reviewer 1 Report
Comments and Suggestions for Authors
This manuscript describes the development of a self-assembling triangular RNA nanoparticle (Bc-triangle) for controlling the fungal pathogen Botrytis cinerea via Spray-Induced Gene Silencing (SIGS). The study is highly innovative, leveraging RNA nanotechnology to address the critical limitations of naked dsRNA, namely poor stability and cellular uptake. The experimental design is logical and comprehensive, spanning from in vitro assays to in planta protection tests. The results demonstrate a clear and significant enhancement in antifungal efficacy and durability compared to linear dsRNA. The topic is of great interest to the fields of plant pathology, RNA biology, and sustainable agriculture. The manuscript is generally well-written, but requires revisions to strengthen the manuscript before it can be considered for publication.
Major Comments:
In Section 3.6, the qRT-PCR analysis only shows data for two (BcDCL1 and BcNMT1) of the four targeted genes. To fully support the claim of "efficient multigene silencing", it is important to present the expression data for all four target genes (BcDCL1, BcPPI10, BcNMT1, BcBAC) following treatment with both Bc-dsRNA and Bc-triangle.
The design and secondary structure prediction of the Bc-triangle are well-described. However, the study lacks experimental validation of the correct folding and three-dimensional structure of the final nanoparticle.
The discussion effectively summarizes the findings but could be improved by a more direct comparison with other nanocarrier systems. Briefly discussing how a pure RNA nanoparticle compares to these synthetic/organic carriers in terms of efficacy, cost, and environmental impact would highlight the novelty of this work.
The manuscript attributes the superior performance of Bc-triangle to "enhanced stability" and "improved cellular uptake," but provides no direct experimental evidence for these claims. It is recommended to incorporate speculation based on existing literature or propose directions for future research into the discussion.
In the title, "enhance" should be corrected to "enhancing"
Line 56:"an d short protection windows" should be corrected to "and short protection windows".
Author Response
|
Comments 1: In Section 3.6, the qRT-PCR analysis only shows data for two (BcDCL1 and BcNMT1) of the four targeted genes. To fully support the claim of "efficient multigene silencing", it is important to present the expression data for all four target genes (BcDCL1, BcPPI10, BcNMT1, BcBAC) following treatment with both Bc-dsRNA and Bc-triangle
|
|
Response 1: [Type your response here and mark your revisions in red] We would like to express my gratitude for highlighting this matter. We concur with this assessment. However, it was observed that BcBAC was the only strain that was identifiable, while PPI10 remained undetectable. Consequently, in the revised manuscript, we have incorporated a discussion on the impact of BcBAC expression following Bc-triangle treatment. The underlying reason for the observed absence of PPI10 expression could be attributed to the specific expression of this gene during the course of pathogen-plant interactions. Consequently, when treating hyphae with RNA nanoparticles, we were unable to detect its expression. In addition, an analysis of the underlying causes of the undetectable PPI10 expression has been incorporated into the Discussion section of the revised manuscript. |
|
Comments 2: The design and secondary structure prediction of the Bc-triangle are well-described. However, the study lacks experimental validation of the correct folding and three-dimensional structure of the final nanoparticle.
|
|
Response 2: Agree. We have added an experiment to characterize the RNA nanostructures by using Cryo-EM in revised manuscript, and the result showed that the Bc-triangle can program self-folding into an expected structure (Please see the supplemental Figure S1). |
|
Comments 3: The discussion effectively summarizes the findings but could be improved by a more direct comparison with other nanocarrier systems. Briefly discussing how a pure RNA nanoparticle compares to these synthetic/organic carriers in terms of efficacy, cost, and environmental impact would highlight the novelty of this work.
|
|
Response 3: Thank you for your comments. We have revised the manuscript to include a comparison of RNA nanoparticles in these aspects within the Discussion section, highlighting the novelty of our study. When compared to synthetic/organic carriers (e.g. layered double hydroxide clay sLDH or chitosan nanoparticles), pure RNA nanoparticles have been demonstrated to exhibit superior stability and RNAi persistence in terms of efficacy, with a significant inhibition of spore germination and mycelial growth. In terms of cost, this process can utilize microorganisms (such as E. coli) for scale-up, thereby eliminating preparation steps involving carrier materials and complexation, significantly reducing costs. With regard to environmental compatibility, pure RNA nanoparticles are fully biodegradable and pose no risk of carrier residue, whereas synthetic carriers may lead to non-target accumulation or ecotoxicity concerns. In comparison to alternative control methodologies, chemical pesticides (e.g., carbendazim) have been observed to readily induce resistance and environmental contamination. Furthermore, genetically modified RNAi crops encounter regulatory challenges in obtaining approval for genetic modification. Conversely, SIGS technology does not necessitate genetic modification and permits flexible application. Additionally, biological agents (e.g., antagonistic bacteria) demonstrate diminished efficacy under environmental constraints, whereas RNA nanoparticles exhibit enhanced stability. In this study, the protective window of Bc-triangle was observed to last up to 10 days, while naked dsRNA was found to last 3–7 days, though this was slightly shorter than the carrier-complexed dsRNA (e.g., sLDH-dsRNA was found to reach 20 days).
|
|
Comments 4: The manuscript attributes the superior performance of Bc-triangle to "enhanced stability" and "improved cellular uptake," but provides no direct experimental evidence for these claims. It is recommended to incorporate speculation based on existing literature or propose directions for future research into the discussion. |
|
Response 4: We are extremely grateful for your thought-provoking inquiry and acknowledge its significance with the utmost seriousness. Although the “enhanced stability” and “improved cellular uptake” properties of Bc-triangle RNA nanostructures have not yet been directly experimentally validated, existing literature allows for reasonable inference: their increased stability may stem from the dense three-dimensional structures formed through self-assembly (such as virusoid capsids or polymer complexes) effectively shielding against nuclease attacks. Improved cellular uptake may be achieved through pathogen-mediated endocytosis or membrane adsorption mechanisms of nanoparticles (particularly given the natural uptake pathway for dsRNA in Botrytis cinerea). Future research should provide direct evidence through the following approaches: comparing degradation kinetics of nanostructures versus free siRNA within mycelium; using fluorescent labeling to track nanostructure localization and internalization efficiency within fungal cells; and elucidating the relationship between surface chemical properties (e.g., charge, ligand modifications) and uptake mechanisms. Existing studies indicate that RNA interference strategies targeting the ergosterol synthesis pathway (e.g., the ERG gene family) in Botrytis cinerea significantly inhibit mycelial growth, while nanocarriers can further optimize delivery efficiency.
|
|
Response to Comments on the Quality of English Language |
|
Point 1: In the title, "enhance" should be corrected to "enhancing" |
|
Response 1: Thank you for your grammatical correction. We have revised the title to “Development of a triangle RNA nanostructure for enhancing RNAi-mediated control of Botrytis cinerea through spray-induced gene silencing without an extra nanocarrier.” |
|
Point 1: Line 56:"an d short protection windows" should be corrected to "and short protection windows". |
|
Response 1: Thank you for your correction. We have replaced “and short protection” in line 56 of the manuscript. |
|
Additional clarifications |
|
All grammatical errors in the English text have been corrected, and information that was unclear has been supplemented or rephrased. |
Reviewer 2 Report
Comments and Suggestions for Authors
Communication: Development of a triangle RNA nanostructure to enhance RNAi-mediated control of Botrytis cinerea via spray-induced gene silencing without an extra nanocarrier
This study examines the effect of exogenous double-stranded RNA (dsRNA) against four B. cinerea virulence genes—BcDCL1, BcPPI10, BcNMT1, and BcBAC. Spray-induced gene silencing (SIGS) technology is quite promising and has been actively developed in recent years. Using SIGS to combat plant pathogenic fungi is a very interesting topic. In my opinion, the relevance of this work is quite high. The text could use some work, specifically removing italics where appropriate, eliminating spaces, and eliminating typos. I highlighted all the errors in yellow in the attached file. I recommend improving the English language. The work is well-planned, but could be improved. Honestly, it remains unclear why the authors assessed the expression of only the BcDCL1 and BcNMT1 genes, although they targeted four other genes. Why wasn't the expression of the BcPPI10 and BcBAC genes analyzed?
There are also a number of methodological questions. I'm still unclear why specifically the BcDCL1, BcPPI10, BcNMT1, and BcBAC genes were used as interference.
What amount of RNA was used for processing? You only provide the concentration, but the volume of solution remains unclear. Please clarify.
The question also remains: why did you use these specific concentrations?
Line 197. What volume of water was used for processing?
Figure 2. Judging by the photograph, the dsRNA was additionally purified from non-specific RNA and DNA. If so, please clarify.
At what stage did you add the RNA to the PDA? Before or after sterilization. Please describe this in more detail.

Author Response
|
Comments 1: This study examines the effect of exogenous double-stranded RNA (dsRNA) against four B. cinerea virulence genes—BcDCL1, BcPPI10, BcNMT1, and BcBAC. Spray-induced gene silencing (SIGS) technology is quite promising and has been actively developed in recent years. Using SIGS to combat plant pathogenic fungi is a very interesting topic. In my opinion, the relevance of this work is quite high. The text could use some work, specifically removing italics where appropriate, eliminating spaces, and eliminating typos. I highlighted all the errors in yellow in the attached file. I recommend improving the English language. The work is well-planned, but could be improved. Honestly, it remains unclear why the authors assessed the expression of only the BcDCL1 and BcNMT1 genes, although they targeted four other genes. Why wasn't the expression of the BcPPI10 and BcBAC genes analyzed?
|
|
Response 1: We would like to express my gratitude for highlighting this matter. We concur with this assessment. However, it was observed that BcBAC was the only strain that was identifiable, while PPI10 remained undetectable. Consequently, in the revised manuscript, we have incorporated a discussion on the impact of BcBAC expression following Bc-triangle treatment. The underlying reason for the observed absence of PPI10 expression could be attributed to the specific expression of this gene during the course of pathogen-plant interactions. Consequently, when treating hyphae with RNA nanoparticles, we were unable to detect its expression. In addition, an analysis of the underlying causes of the undetectable PPI10 expression has been incorporated into the Discussion section of the revised manuscript.
|
|
Comments 2: There are also a number of methodological questions. I'm still unclear why specifically the BcDCL1, BcPPI10, BcNMT1, and BcBAC genes were used as interference.
|
|
Response 2: Agree. In earlier research, it was established that tomato-encoded sRNAs have the capacity to cross-target genes that are encoded by Botrytis cinerea. Furthermore, tomato sRNAs that target pathogenic genes have been observed to exhibit a significant antibacterial effect against Botrytis cinerea. Consequently, the present study randomly selected four pathogenic genes from the ten previously reported pathogenic genes for multifunctional RNA nanoparticle design and antibacterial activity analysis. In the final paragraph of the revised manuscript's introduction, the rationale for selecting these four pathogenicity genes has been incorporated, as described above.
|
|
Comments 3: What amount of RNA was used for processing? You only provide the concentration, but the volume of solution remains unclear. Please clarify.
|
|
Response 3: We would like to express my gratitude for the question you have posed. We extend our sincere apologies for any misunderstanding that may have arisen due to the ambiguity in our initial communication. In order to achieve the desired concentrations, 7.5 mL of RNA solution was added to each 150 mL of PDA medium, resulting in final working concentrations of 25, 50, 100, and 200 ng/μL. The control group, which did not receive any treatment, received an equal volume of ddH2O. In relation to the aforementioned issue, we have made revisions in Section 2.5 of the Methods section in the manuscript.
|
|
Comments 4: The question also remains: why did you use these specific concentrations?
|
|
Response 4: The author wishes to express their gratitude for the thoroughness with which the issue was addressed. The concentration setting was determined through a comprehensive review of existing scholarly precedents and a meticulous analysis of our own group's prior research. A comprehensive review of the extant literature revealed that, prior to the present study, working concentrations of 100~300 ng/μL of RNA had been identified as a general effective solution for gene silencing. Consequently, we have established 100 ng/μL as the definitive working concentration for the spore germination inhibition experiment, with the objective of preliminarily assessing the inhibitory effects of our designed RNA against gray mold. Subsequently, to more accurately determine the optimal concentration for plant spraying, a concentration gradient experiment was conducted on plates. The results analysis in Section 3.5 indicates that a working concentration of 200 ng/μL was identified as optimal. This concentration was ultimately adopted for the plant spraying experiment.
|
|
Comments 5: Line 197. What volume of water was used for processing?
|
|
Response 5: In order to address this issue, an equal volume of ddH2O was added to the RNA solution of the experimental group, specifically 7.5 mL of ddH2O per 150 mL of medium. It should be noted that Section 2.5 of the method has been modified; refer to the manuscript for further details.
|
|
Comments 6: Figure 2. Judging by the photograph, the dsRNA was additionally purified from non-specific RNA and DNA. If so, please clarify.
|
|
Response 6: The dsRNA and Bc-triangle were extracted derectly from IPTG-induced E. coli using a dsRNA extraction kit (RNA-Direct Silencing Technology Co. Ltd, AK011, Shaoxing, China).In order to elucidate the results, the undiluted crude extract was subjected to agarose gel electrophoresis analysis. In the revised manuscript, we have added the preparation method of dsRNA and Bc-triangle in the ‘Material and Methods’ section.
|
|
Comments 7: At what stage did you add the RNA to the PDA? Before or after sterilization. Please describe this in more detail. |
|
Response 7: Thank you for asking this question. Here's our response: After PDA sterilization (121°C, 15 min), cool to 50~60℃. Then add the RNA solution calculated based on the final working concentration. Mix thoroughly, and pour into petri dishes. We have also updated the content in Section 2.7 of the revised manuscript.
|
|
Additional clarifications |
|
All grammatical errors in the English text have been corrected, and information that was unclear has been supplemented or rephrased. |
Round 2
Reviewer 2 Report
Comments and Suggestions for Authors
Thanks for clarifying my questions. This version suits me.